# First, Be a Good Citizen: Organizational Citizenship Behaviors, Well-Being at Work and the Moderating Role of Leadership Styles

**DOI:** 10.3390/bs13100811

**Published:** 2023-09-30

**Authors:** Reinaldo Sousa Santos, Eva Petiz Lousã, Maria Manuel Sá, João Alves Cordeiro

**Affiliations:** 1Research Unit in Business Sciences and Sustainability (UNICES), University of Maia, 4475-690 Maia, Portugal; d011870@umaia.pt (E.P.L.); maria.sa@umaia.pt (M.M.S.); 2Centre for Organizational and Social Studies of Polytechnic of Porto (CEOS.PP), Polytechnic of Porto, 4465-004 Porto, Portugal; 3NECE-UBI, Research Centre for Business Sciences, 6200-209 Covilhã, Portugal; 4Department of Business Sciences, University of Maia, 4475-690 Maia, Portugal; a038211@ismai.pt

**Keywords:** organizational citizenship behavior, well-being at work, leadership styles, human resource management

## Abstract

The study investigates the effect of organizational citizenship behavior (OCB) on well-being at work. The study further examines the moderating role of people and task-focused leadership styles between OCB on well-being at work. Individual-directed organizational citizenship behaviors (OCBI) and organizational-directed organizational citizenship behaviors (OCBO) will also be analyzed. A quantitative study was conducted and convenient sampling was adopted in selecting respondent workers (*n* = 200) in different Portuguese organizations. The results show that OCBs positively and significantly influence well-being at work. The strength of individual-directed organizational citizenship behaviors (OCBI) on well-being at work is stronger than that of organization-directed organizational citizenship behaviors (OCBO). Contrary to expectations, the relationship between leadership styles and well-being was not statistically significant, offering possibilities for discussion regarding the central importance usually attributed to leadership in the organizational context. However, leadership styles have a moderating effect between OCB and well-being at work, except when the employee adopts OCBO and the leadership style is people-oriented. The present study is innovative because it positions OCB as an antecedent in the relationship with well-being at work and investigates the moderating role of leadership styles in the relationship between organizational citizenship behavior and well-being.

## 1. Introduction

The principles of sustainability and social responsibility advocate that organizations should act to protect and value their stakeholders, in a medium- and long-term perspective [1], paying close attention to well-being at work [2,3,4]. The importance of well-being will increase as a lasting balance point in the employer–employee relationship [5]. As a result, positive and productive work environments are associated with well-being at work (e.g., [6,7,8,9,10]) and valuing the ethics and social conformity of the behaviors of employees and leaders [11]. Leaders need to reconcile concern for performance with the ability to support and motivate people, ensuring properly sized, qualified and motivated teams [12]. Valuing well-being at work is an informed commitment by organizations with a view to increasing their levels of efficiency [13]. At the same time, providing well-being is an ethical duty [14] because work is a fundamental dimension of current life in society and, for people to have good lives, they need to have good jobs and work in good organizations [15]. In a context characterized by the reinforcement of competitiveness, efficiency [16], turbulence and ambiguity [17], the high performance of organizations requires that employees deliver a performance that goes beyond the execution of prescribed tasks only [18]. Organizational citizenship behaviors, characterized by their ethical and voluntary nature, promote improved contextual performance [19] and increased well-being at work [20] among other organizational outcomes.

However, the literature has given more importance to leadership than to organizational citizenship as a source of several positive results at work, namely in terms of well-being at work. The present study intends to evaluate the relationship between OCB, leadership and well-being at work, considering OCB as an antecedent variable, and answering the following research questions: (i) Do OCBs promote well-being at work?; (ii) Do leadership styles promote well-being at work and, if so, which ones?; and (iii) Is the effect of OCBs on well-being at work conditioned by leadership styles? This is an innovative approach regarding the positioning of the variables in the research model, which proved to be correct given the results obtained in the present study.

Thus, the overall objective of this study is to analyze the impact of OCB on well-being at work and investigate the moderating role of leadership styles in OCB and well-being at work. Specifically, we seek to analyze:(a)the relationship between OCB and well-being at work;(b)the relationship between leadership styles and well-being at work;(c)the moderating effect of leadership styles on OCB and well-being at work.

In the following sections, we present the theoretical background and the hypothesis development, followed by materials and methods, results and discussions. Finally, theoretical and practical contributions, limitations and future research are presented.

## 2. Theoretical Background and Hypothesis Development

### 2.1. Organizational Citizenship Behaviors

Organizations prefer employees who, going beyond established contractual and functional requirements, act in favor of valuing the psychological and social environment at work [21] and invest in building a collective framework that improves the performance of tasks and life in common [22]. These behaviors are called organizational citizenship behaviors (OCB) [23,24] and add a diverse set of constructs present in the organizational literature [25], namely prosocial organizational behaviors [26], the work–organizational spontaneity relationship [27], extrarole behaviors [28] and contextual performance [19]. OCBs are discretionary, individual and voluntary behaviors that promote the improvement of the organization’s performance and are not recognized by the current formal reward system [29,30]. Although they consist of an altruistic performance [31], OCBs increase the employee’s sense of importance [32], generate self-compensation [33] and can also reinforce rewards through favorable performance evaluations [34]. They are extrafunctional behaviors [35,36], specific and deliberate, intended to support coworkers and organizational success [37,38,39]. OCBs are positively related to several organizational outcomes, namely service quality and healthy organizational climate [40], productivity, performance and collaboration [36,41,42], creativity [36,43,44], automation and a sense of belonging [45], well-being at work [20], leadership competence [30,46], human resource management systems [47,48], the work–family spillover effect [49], organizational efficiency [50] and the competitive advantage of organizations [51]. Many authors consider OCB a personality trait [52], while others consider it a latent construct or a standard of ethical behavior at work that survives the most adverse contexts [53,54]. The voluntary action of helping others stems from an ethical individual impetus [55], to the extent that this “good soldier” behavior [56] is not part of the formal functions of those who help [57,58], nor is there a penalty for those who do not do so [30,59].

OCB includes five types of behavior considered important for reinforcing organizational efficiency and well-being at work [21]:Altruism refers to the behaviors assumed by employees in specific situations that often involve helping voluntarily in the fulfillment of a task, without expectation of reward or compensation [60]. In a more global sense, altruistic behavior contributes to making others happier [61]. It includes, for example, supporting employees who are overloaded with work or guiding them in more difficult or complex tasks [62].Conscientiousness refers to the pattern of behaviors that generate a healthy and consistent work environment and exist even when the employee is not observed. Conscientious behavior goes beyond the minimum requirements established by the organization and general compliance and can be defined as “professional pride and care” [63]. It includes, for example, being punctual, being present at meetings, keeping the workstation tidy and not abusing breaks at work [21,64]. The conscientious employee is more responsible and needs less supervision [65].Sportsmanship refers to how much employees are willing to accept less pleasant situations in favor of the “greater good”. It consists of adopting a positive attitude in the face of adversity and being tolerant of the organization’s problems and difficulties [66,67,68]. It includes, for example, not openly criticizing colleagues or being willing to maintain one’s commitment even under uncomfortable working conditions [21,25].Courtesy refers to behaviors that are oriented towards the protection of a healthy and positive work environment. It consists of showing respect for others and avoiding the emergence of interpersonal problems and conflicts [61,67]. It includes, for example, warning a colleague not to be late or consulting other people before making a decision that may affect them [53,65,68].Civic virtue refers to the level of readiness and interest shown in actively participating in the political life of the organization [69,70,71]. Employees with civic virtue are actively involved in issues relevant to the organization and contribute constructively, innovatively and creatively [61,72] and are concerned that the organization has a positive image and status [73]. It includes, for example, handling incoming mail, carefully reading notices and posted information and participating in debates with free and frank opinions [21].

The five dimensions of OCB can also be aggregated into individual-directed organizational citizenship behaviors (OCBI)—including the dimensions of altruism and courtesy—and organizational-directed organizational citizenship behaviors (OCBO)—including the dimensions of sportsmanship, civic virtue and conscientiousness [74].

### 2.2. Well-Being at Work

Work can offer pleasant activities, structured occupation for the hours of the day, promote social interaction and generate opportunities for involvement, challenge and meaning [75]. Being employed has a very positive impact on people’s well-being [4], as demonstrated by the fact that unemployed people have less well-being than employed people [76]. Work satisfies many of people’s most critical needs, whether to collect tangible resources to respond to vital and security needs, or to develop skills, realize one’s purpose and integrate into an active social network [77]. In the same way that subjective well-being refers to the individual and general evaluation that a person makes about his or her own life, as a result of having experienced more positive emotions than negative emotions [78,79,80], well-being at work also refers to an individual and general assessment of the quality of the work experience [81]. Well-being at work is more than “not being sick at work” [82] (p. 55) and implies that the employee obtains pleasure and fulfillment through work [83,84]. Well-being at work translates into a state of consciousness that includes ‘cold cognitions’, such as evaluative beliefs and judgments, and ‘warm affective phenomena’, such as mood and emotions [10,85]. Well-being at work is characterized by three dimensions [86]: psychological, physical and social:Psychological well-being refers to the pleasure and fulfillment obtained, holistically, from doing work. It responds to the hedonistic and eudemonic appeals of human beings and considers well-being at work as a “global sensation of pleasure laden with meaning”, says Ben-Shahar [87] (p. 72). The hedonic experience includes positive affect, as joy and enthusiasm, and the absence of negative affect, such as a lack of anxiety and a sense of calm [6]. Experiencing positive emotions is crucial for the employee to thrive, mentally and psychologically, on and off the job [9]. The eudemonic experience transcends the mere hedonistic pleasure of the moment [88] and is associated with the realization of the individual’s potential and the search for meaning and purpose at work [14]. Those who are closer to their daimon—“true self”—have more well-being, because well-being is associated with self-fulfillment and the possibility of taking advantage of opportunities for individual appreciation and growth [89,90].Physical well-being refers to the impact of work on the employee’s health and includes situations of injury, illness and risk of stress [86]. The reinforcement of the flexibility and precariousness of work relationships, as well as the intensification of work circuits and demands [4,91,92], add to the usual physical sources of injuries and illnesses, increasing situations of stress, harassment and violence at work [77,93]. Well-being is negatively correlated with depression, anxiety and burnout and positively correlated with physical health [94].Social well-being refers to the quality of social relationships at work [89], including short-term interactions and long-term relationships with others [95]. Social relationships respond to each individual’s need to belong and have a positive impact on their levels of energy, self-awareness and support [96,97]. Employees have higher job satisfaction and positive affect on days when they experience more positive social interactions, and report lower job satisfaction and higher negative affect when they experience more negative interactions [96,98]. Positive relationships are characterized by trust, respect, loyalty and a sense of mutuality [89,99,100] and have a positive impact on strengthening the employees’ instrumental and emotional capacity, as well as improving their possibilities for career progression and expanding their circle of friends [97]. People assess the coherence of the perceived reality at work with the values announced by the organization [101,102] and with the values they consider fundamental for their well-being at work [103]. Social well-being is strongly affected by this ethical scrutiny that employees permanently carry out at work [104].

Employees act according to the general perception they have of their work situation [94], which attributes to the characteristics of the work—tasks, leadership, rewards, social and ethical environment—a decisive importance for valuing well-being at work and organizational efficiency [77].

### 2.3. Leadership Styles

The organizational literature is replete with different definitions of leadership [105]. A leader sets the direction, aligns, motivates and inspires people in their teams [106] to contribute positively to the objectives of the organization to which they belong [107]. Leadership is the process, formal or informal, in which a person interacts and influences a group of people, in order to establish and maintain performance standards, create and develop specific skills for the job and, globally, achieve defined optimal results. for the group, making use of the company’s resources in a thoughtful and coherent way [108,109]. Leadership is based on the responsibility of directing the efforts of a group of people towards a common goal [110]. Leaders have a positive impact when they are a source of instrumental and emotional support for their team members and have a negative impact when they amplify the sources of stress at work, through role ambiguity and poor distribution of work resources and recognition [111]. Destructive leadership has a high potential to reduce levels of health, performance and well-being [112,113]. Effective leadership depends on the leader’s perception of the situation, the constraints that frame it, and their ability to act in an adjusted and properly contextualized way [114]. There is no single type of leadership that can be successfully applied in all situations, requiring the leader to be able to read the situation and adapt the leadership style to the contingent reality in which he operates [115].

The organizational literature is rich in the presentation of different styles of leadership, characterizing them as repetitive practices dedicated to building relationships, gathering information and making decisions [116]. The behavioral approach considers that leaders differ according to their orientation towards the task or towards people. In alignment, the situational model [117] characterizes the leader through the combination of directive behavior (focus on the execution of tasks) with support behavior (focus on the development and satisfaction of people). Leaders with a focus on tasks demonstrate great concern with the quality of performance and with achieving and exceeding defined objectives and tend to provide individual feedback and seek short-term goals. Leaders with a focus on people show great concern for the quality of interpersonal relationships, for individual well-being and for creating and protecting team spirit. Greater concern for people leads them to promote autonomy and participation and tending to seek medium and long-term goals, for which they consider it essential to have well-qualified and motivated teams. The literature on leadership styles also uses the task–people dualism in determining their various typologies. The autocratic and bureaucratic styles focus on the task, while the democratic and charismatic styles value the quality of relationships with people [118,119]. In the last three decades, much research has emerged on transactional and transformational leadership [120], making it possible to clarify that transactional leadership focuses on exchanging resources, by offering extrinsic rewards in return for employee effort and obedience, while transformational leadership focuses on valuing the competence and autonomy of employees, in a long-term view that pays greater attention to individual well-being and the sustainability of organizational action [121,122,123]. Transactional leadership styles value contingent rewards and high supervision and control [124] and transformational leadership styles impose an active leader involvement through charisma and idealized influence, inspirational motivation, intellectual stimulation and personal and individual attention [125], with more positive results in performance, organizational commitment, well-being at work and OCB than transactional styles [30,126,127,128,129]. Transactional leadership prioritizes tasks and transformational leadership values people more. Responsible [130], authentic [131] and servant [132] leadership are also presented through attributing more to people to the detriment of a more task-centered and results-oriented approach. short term. In Goleman’s [109] typology, the visionary, affiliative democratic and coaching styles show high emotional intelligence, have a positive impact on the team’s climate, and therefore show greater concern for people. The commanding and pacesetting styles, which value decision speed, excellent performance and overcoming objectives, tend to increase conflict and silence behaviors [133,134], thus demonstrating greater concern with the tasks. These styles reduce organizational constraints at work [111], understood as the organizational obstacles that prevent employees from realizing their full potential and delivering a high level of performance at work [135]. Goleman [109] argues that people-oriented styles are more effective, as they positively influence the organizational climate, and a positive climate has a direct correlation with various organizational results, such as sales profit, growth in the volume of business, efficiency and profitability. Later studies have confirmed this effectiveness in different activity sectors and organizational contexts (e.g., [111,136,137,138,139]). People prefer leaders who have a positive impact on the group climate and are willing to return, in reciprocity [140], more quality in work and relationships [141,142,143].

### 2.4. Organizational Citizenship Behaviors and Well-Being at Work

OCBs refer to cooperative behaviors with peers, performing additional tasks without protest, protecting the organization’s resources, the efficient use of time, sharing information, knowledge and ideas and any other behavior that favorably represents the organization [54]. The search for extrafunctional behaviors arises when affective bonds are developed with the organization, either because the individual–organization relationship is perceived as valuable or because the individual identifies with the organization’s philosophy [144]. The literature has focused on showing that well-being at work increases OCBs (e.g., [145,146,147,148]), but, considering that OCBs occur as a result of eminently moral motivations [57,145], it is possible to perceive them upstream of well-being at work and not just as its consequence.

According to the JD-R model, a high level of well-being, even in more adverse contexts, reduces the level of burnout and stress and increases people’s levels of commitment and involvement at work [149,150]. Bakker and Demerouti [149] consider that well-being allows balancing the existing interaction between the physical, mental and emotional demands of work and the employee’s resources to carry it out, including autonomy, social support or feedback. According to the constellation of demand, when the employee has high resources, tension decreases and motivation and well-being increase at work. OCB can be assumed as a job resource, insofar as it responds to the authors’ definition: “job resources refer to those physical, psychological, social, or organizational aspects of the job that are either/or: functional in achieving work goals; reduce job demands and the associated physiological and psychological costs; stimulate personal growth, learning, and development.” [149] (p. 312). OCB, as a personality trait [52], a latent construct or a standard of ethical behavior at work [53,54], is a psychological resource that individuals possess at different levels and that, in combination with other resources and job requirements, generates well-being at work. Additionally, if the JD-R model considers the constructs of social support, performance feedback and organizational commitment as work resources with an impact on well-being (e.g., [149,151,152]), it is natural that OCB, whose nature includes similar behavior, may also occupy a similar position in this relationship of influence towards well-being at work. The placement of OCB upstream of well-being at work does not contradict the literature that positions it downstream of well-being at work, because positive concepts at work tend to generate reciprocity mechanisms [138] in which they feed each other. As Fisher observes [85], there is common core of happiness across the different constructs at work. “Giving to others” is one of the dimensions of positive relationships at work and a source of well-being at work [97], namely because this generosity of behavior generates expectations of mutuality in relationships at work [100]. Also the dark side of OCB [153] has a negative impact on well-being [154]. OCBs are deviant when behavior violates organizational norms, threatening the well-being as a whole and its individual members [155]. In fact, in “The Job Demands-Resources model: state of the art”, Bakker and Demerouti [149] dedicate a section to the reciprocity relationships existing between different organizational constructs, pointing out that the literature has shown significant results regardless of the positioning that the variables occupy in the relationship and many studies with inverted causality have contributed with relevant knowledge. For example, the authors claim that “job stress and motivation can both be outcomes as well as predictors of job demands and resources” [149] (p. 321). More recently, the authors [156] warn that the intensification of positive behaviors at work, namely OCB, can create a spiral of gains with positive impacts on several organizational variables. This understanding can be applied to the existing relationship between well-being at work and OCB, although the literature has only dedicated attention to placing OCB as an effect of well-being at work. This fact grants even greater relevance and novelty to the analysis model designed in this research project.

In the present study, we adopted the well-being perspective at work, which comprises both affective (emotions and moods) and cognitive (perceived achievement) aspects. Well-being at work includes positive experiences. Thus, when there is well-being, positive affect at work prevails over negative affect, and workers experience personal fulfilment by developing their potential [157]. It is expected that OCBs will increase well-being at work, sometimes people-oriented (OCBI) and sometimes organization-oriented (OCBO). Accordingly, the following hypotheses were defined:

**Hypothesis** **1 (H1).**
*There is a positive relationship between OCB and well-being at work.*


**Hypothesis** **1a (H1a).**
*There is a positive relationship between OCBI and well-being at work.*


**Hypothesis** **1b (H1b).**
*There is a positive relationship between OCBO and well-being at work.*


### 2.5. Leadership Styles and Well-Being at Work

Leadership style refers to the ability of leaders to monitor personal and interpersonal emotions, discriminate them and use them to regulate and guide thoughts and actions [158]. The leader’s situational and behavioral decision tends to give greater priority to the efficiency of results or the well-being of people [117]. Leaders who demonstrate a broader domain of leadership styles [108] can establish an emotional rapport with their teams and tend to be more effective [109]. The leadership styles that demonstrate the greatest concern for people—encouraging interpersonal relationships, teamwork and collaboration, and prioritizing employee wellbeing—tend to have a positive impact on well-being at work, while leaders who adopt task-oriented leadership style—focusing on completing necessary tasks to reach organizational targets, prioritizing performance and short-term goals rather than focusing on employees—tend to penalize well-being at work. However, people-focused leadership styles can also penalize individual and organizational performance and, as a result, become detrimental to well-being at work. The performance penalty has a negative effect on well-being at work [91], justifying, for example, the value placed on technical and cognitive planning, decision-making and problem-solving skills in leadership profiles [159], while their absence characterizes destructive leadership [112,113]. Effective leadership combines functional authority skills with the ability to develop positive interpersonal relationships, sharing empathy and trust with the team [160]. Accordingly, the following hypotheses were defined:

**Hypothesis** **2 (H2).**
*There is a positive relationship between leadership and well-being at work.*


**Hypothesis** **2a (H2a).**
*There is a positive relationship between a leadership focus on people and well-being at work.*


**Hypothesis** **2b (H2b).**
*There is a positive relationship between a leadership focus on results and well-being at work.*


### 2.6. The Moderating Effect of Leadership Styles on Organizational Citizenship Behaviors and Well-Being at Work

Leadership is the visible face of the organization; it conveys the organizational culture and establishes the benchmark for desirable behavior or accepted performance for team members [161]. The OCB, that individual ethical impetus that drives employees to help their colleagues (OCBI) or the functioning of the organization (OCBO), can be amplified or decimated by the style adopted by the leader. As with the relationship between OCB and well-being at work, the literature has also analyzed the relationship between OCB and leadership, positioning OCB downstream of leadership, although with inconsistent results (e.g., [162,163,164,165,166,167]). When looking for a moderating effect, OCB tends not to occupy the position of an independent variable (e.g., [168,169]). However, if OCB represents an ethical impulse [53,55], and if individual ethics refer to the ways in which subjects position themselves in the social contexts in which they operate [170], then it is justified to position OCB as an independent variable and assess how important leadership styles may be in determining, and assuming a moderating effect on the relationship between OCB and well-being at work. This evaluation will analyze this moderating effect considering the various dimensions of OCB and leadership. Accordingly, the following hypotheses were defined:

**Hypothesis** **3 (H3).**
*The positive relationship between OCB and well-being at work will be stronger when leadership is high.*


**Hypothesis** **3a (H3a).**
*The positive relationship between OCB and well-being at work will be stronger when leadership focus on tasks is high.*


**Hypothesis** **3b (H3b).**
*The positive relationship between OCB and well-being at work will be stronger when leadership focus on people is high.*


**Hypothesis** **3c (H3c).**
*The positive relationship between OCBI and well-being at work will be stronger when leadership focus on tasks is high.*


**Hypothesis** **3d (H3d).**
*The positive relationship between OCBI and well-being at work will be stronger when leadership focus on people is high.*


**Hypothesis** **3e (H3e).**
*The positive relationship between OCBO and well-being at work will be stronger when leadership focus on tasks is high.*


**Hypothesis** **3f (H3f).**
*The positive relationship between OCBO and well-being at work will be stronger when leadership focus on people is high.*


The theoretical model of this study is depicted in Figure 1.

## 3. Materials and Methods

### 3.1. Participants

For this study, we used a nonprobabilistic sampling technique using convenience sampling, where participants were selected accidentally and voluntarily. The target population was considered the active population over 18 years of age who performed functions in the Portuguese territory and were in contact with colleagues and hierarchical superiors. The sample was composed of 200 workers from different Portuguese organizations (cf. Table 1). The majority of respondents in our sample were female (82%), and the age of the workers varied between 18 and 66 years (M = 40.02 years, SD = 11.67 years). Regarding their level of education, 28% completed primary or secondary education, while 72% continued studies, having attended higher education or technical training, distinct from professional courses with equivalence to secondary education.

### 3.2. Measures

Leadership styles hetero-assessment scale: This construct was measured using a 15-item scale developed by Mourão et al. [171], which captures two dimensions of leadership (focus on people and focus on tasks). The scale was constructed from interview questions and focus groups with people with different characteristics such as sex, age, and professional areas. A sample item for the dimension focus on people is: “Encourages the personal development of your team”. Internal consistency was α = 0.90. A sample item for the dimension focus on tasks is: “Is very concerned about the fulfillment of the work tasks”. Internal consistency was α = 0.82. Respondents were asked to agree or disagree with each item (1 = strongly disagree and 10 = strongly agree). The scale was adapted to Portugal, adapting linguistic corrections, and we obtained a bifactorial structure explaining 81.6% of the total variance of the instrument. The internal consistency of the factor focus on people was α = 0.96 and of the focus on tasks was α = 0.84. The final scale validated in the present study consists of 9 items (6 items from the focus on people factor and three items from the focus on tasks factor).

Organizational citizenship behavior: This construct was measured using a 19-item scale developed by Konovsky and Organ [172] and adapted and translated to Portuguese by Santos [167]. The concept of OCB comprises five dimensions (altruism, courtesy, sportsmanship, civic virtue, and conscientiousness). The adaptation of Santos [173] becomes relevant for the present study because it ensured the translation and passage of items from the third to the first person, positioning as statements from the perspective of the worker (e.g., “In my organization, I help others who have heavy workloads”). The items of this construct were measured on a seven-point Likert scale, ranging from (1) “strongly disagree” to (7) “strongly agree”. The Cronbach’s reliability of the instrument in this study was 0.918. The five dimensions of OCB were aggregated into individual-directed organizational citizenship behaviors (OCBI)—including the dimensions of altruism and courtesy—and into organization-directed organizational citizenship behaviors (OCBO)—including the dimensions of sportsmanship, civic virtue, and conscientiousness [72]. The Cronbach’s reliability of OCBI and OCBO were 0.961 and 0.756, focus on tasks.

Work well-being scale: This construct was measured using a 30-item scale developed by Paschoal and Tamayo [157]. The scale is composed of three dimensions (positive affect, negative affect, and expressiveness and achievement). A sample item is, “I develop skills that I consider important”. The items of this construct were measured on a 5-point scale. For the affect dimension, possibilities of responses were represented from (1) “Not little” to (5) “Extremely”. In the dimensions expressiveness and achievement at work, the values assumed the meaning of (1) “strongly disagree” to (5) “strongly agree”. The reliability coefficients of the original scale ranged from 0.88 to 0.93, and in the present study, they ranged from 0.91 to 0.94.

### 3.3. Procedures

Data were collected at a single point in time through online self-administered questionnaires between February and March 2022. The data were gathered using a Google Forms questionnaire distributed via a link in various social networks (e.g., Facebook, Whatsapp, and LinkedIn), which included details about the project goals, a request to express their consent to participate in the study, and the questionnaire. Participants were informed that both confidentiality and anonymity were assured. Previously, we conducted a questionnaire pretest on seven workers from various areas and academic backgrounds to validate the content and understanding of the questions. We made some linguistic adjustments in the original instrument’s language (Brazilian Portuguese) and the language used for the current study’s respondents based on the pretest participants’ suggestions (Portuguese of Portugal).

### 3.4. Data Analysis

All statistical analyses were conducted using IBM SPSS Statistics software (version 28). An exploratory factor analysis was performed on the leadership styles hetero-assessment and work well-being scales in order to validate for Portugal. Cronbach’s alpha was used to assess the item’s internal consistency and, thus, the reliability of the research instruments. The statistical procedure began with the determination of descriptive statistical measures such as mean, standard deviation, maximum and minimum of all variables observed in the questionnaires. Student’s *t*-test and ANOVA F-test were then used to compare key variables of the study (Leadership, OCB, and Well-Being) by gender, age groups and education level. Pearson’s correlation coefficients were subsequently determined to analyze the relationship between these variables. A simple moderation model 1 was proposed, using the Process macro v.4.2 developed by Andrew Hayes [174] to assess the moderation effect.

## 4. Results

As a first step, Student’s *t*-test and ANOVA F-test were used to compare key variables of the study (Leadership, OCB, and Well-Being) by gender, age groups and education level, without significant differences. Then, the descriptive statistical measures (mean, standard deviation, minimum and maximum) were calculated, and the Pearson’s correlation coefficients were determined to analyze the relationship between the scales and subscales (Table 2).

Regarding the correlation results, there was a significant positive correlation between OCB and WWB (work well-being) (r = 0.51; *p* < 0.001), supporting H1. There was also a significant positive correlation between the dimensions of OCBI and WWB (r = 0.54; *p* < 0.001) and OCBO and WWB (r = 0.38; *p* < 0.001), supporting H1a and H1b, respectively. Analyzing the strength of the relationship between OCB and the two subdimensions (OCBI and OCBO) with employees’ well-being, we found that the relationship with OCBI is the strongest. Additionally, there was no correlation between neither of the dimensions of leadership and the variable WWB. Therefore, the hypotheses H2, H2a and H2b were not supported. It is also observed that there is no correlation between leadership and OCB or its dimensions.

### Testing Moderation Hypothesis

The testing of the moderation model using the Process macro revealed that leadership (L) has a moderating effect on the relationship between OCB and WWB (R^2^ = 0.33, F(3,196) = 31.48, *p* < 0.0001) The interaction between OCB and leadership showed a statistically significant effect indicating the presence of moderation ($R_{Change}^{2}$ = 0.05, F(1,196) = 14.30, *p* < 0.001) supporting H3 (Table 3). The same is true when considering one of the leadership dimensions as the moderator variable: LFT as a moderator variable: (R^2^ = 0.33, F(3,196) = 31.70, *p* < 0.0001), with the interaction between OCB and LFT ($R_{Change}^{2}$ = 0.05, F(1,196) = 14.20, *p* < 0.001), supporting H3a; and with LFP as the moderator variable (R^2^ = 0.31, F(3,196) = 29.19 *p* < 0.0001), with the interaction between OCB and LFP ($R_{Change}^{2}$ = 0.03, F(1,196) = 14.20, *p* = 0.01), supporting H3b (Table 3). Considering OCBI as predictor variable (X) of WWB (Y) and one of the dimensions of leadership as a moderator variable (W), there is also moderation: LFT as moderator variable (R^2^ = 0.34, F(3,196) = 33.41 *p* < 0.0001), with the interaction between OCBI and LFT ($R_{Change}^{2}$ = 0.05, F(1,196) = 13.70, *p* < 0.001), supporting H3c; and LFP as a moderator variable (R^2^ = 0.34, F(3,196) = 33.37, *p* < 0.0001), with the interaction between OCBI and LFP ($R_{Change}^{2}$ = 0.05, F(1,196) = 14.13, *p* < 0.001), supporting H3d (Table 3). However, when OCBO is considered as a predictor variable and one of the leadership dimensions as a moderator, there is presence of moderation only when LFT is considered as a moderator variable (R^2^ = 0.19, F(3,196) = 15.18, *p* < 0.0001), with the interaction between OCBI e LFT ($R_{Change}^{2}$ = 0.04, F(1,196) = 9.45, *p* < 0.001), supporting H3e. Thus, hypothesis H3f was not supported (Table 3).

The moderator variable (W) was split into three parts to understand the moderation effect better, adopting the cut-off points relating to the 16th, 50th and 84th percentiles [175]. This procedure was carried out on all models in Table 3, except the last model, since moderation is absent. The results of all three cut-off points were statistically significant. As the value of the moderator variable (L, LFT or LFP) increases, the impact of OCB on WWB also increases. Given the example of Model 1, where OCB is the independent variable (X), WWB is the dependent variable (Y), and leadership is the moderator variable (W), we found that when the level of leadership is low, the relationship between OCB and well-being is positive (β = 0.118 *p* < 0.001). For intermediate levels of leadership, the relationship remains positive but becomes stronger (β = 0.226 *p* < 0.0001) and for higher levels of leadership, the relationship remains positive but stronger (β = 0.287 *p* < 0.0001). In other words, the higher the leadership level, the stronger the impact of OCB on well-being. To better visualize this conditional effect of the focal predictor in Model 1, we present Figure 2.

This same effect can be visualized for the remaining models in Table 3 and in Figure 3, Figure 4, Figure 5, Figure 6 and Figure 7.

## 5. Discussion and Conclusions

This study aimed to deepen knowledge about the existing relationships between three very relevant concepts in organizational literature—OCB, leadership and well-being at work—based on an innovative framework that positions OCB as an antecedent variable. Looking at work relationships from the premise “First, be a good citizen”, it was possible to collect relevant information to deepen current knowledge about the aforementioned constructs and about how organizational management can reinforce its levels of efficiency and sustainability.

First, our results confirm that OCBs have a positive impact on well-being at work, but the impact is more robust when OCBs are people-oriented (OCBI) than when they are organization-oriented (OCBO). Employees feel better when they adopt the behaviors of altruism and courtesy than when they adopt the behaviors of sportsmanship, civic virtue and conscientiousness. The main effect of adopting OCBOs may be, instead of reinforcing well-being, the reinforcement of the organization’s efficiency. It was said that people who adopt extrafunctional and prosocial behaviors in favor of the organization and their coworkers are reinforcing their own well-being at work. The theory of social exchange advocates that social relationships involve a set of interactions that generate obligations, seen as interdependent and contingent on the actions of other people [140,175] and that these interactions have the potential to generate greater quality and return for all parties involved in the relationship [176]. Contingent reactions to each other’s actions provide mutually rewarding transactions and relationships [176,177]. Those who benefit from support at work will tend to respond accordingly, fueling a virtuous circle of mutuality based on social support and cooperation [86]. OCBs increase well-being at work and well-being increases OCBs. More important than determining where to position the “egg” and the “chicken” in this relationship, it is essential to ensure that the work environment and experience promote them, in favor of strengthening the sustainability of the business and life at work [178].

Second, our results show that well-being at work is more associated with shared citizenship behaviors than with leaders’ style of action. It was not possible to identify a relationship between leadership and well-being at work as the literature often suggests (e.g., [179,180,181]), which curbs the importance attributed to leaders as agents capable of decisively influencing the entire organizational reality. Relationships at work are also influenced by nonexclusively professional frameworks [182] and the cultural framework influences the style and importance of leadership for organizational results [183]. Our study was carried out in Portugal, whose culture is characterized by “high power-distance, mildly collectivist, feminine and strongly avoidant uncertainty” [184] (p. 185), at the antipodes of the American cultural model [185], in which leadership tends to acquire greater importance in organizational reality. In Portugal, leaders tend to adopt a leadership style that ensures employee protection [186] and employees tend to value informal interactions with leaders [185]. Even in this cultural context, it was not possible to show the association between people-oriented leadership and well-being at work, reinforcing the idea that there are other organizational variables with greater capacity to generate well-being at work, namely adoption of OCBs.

Finally, and although leadership has no direct impact on well-being at work, our results show that it has a positive moderating effect on the relationship between OCB and well-being at work, except when OCBs are organization-oriented and the leadership style is people-oriented. When the employee mobilizes to help the organization by adopting sportsmanship, civic virtue and conscientious behaviors and envisions a leadership style that is more concerned with people, there is a misalignment of action priorities that can justify their noncontribution to the improvement of well-being at work. Essentially, leadership styles amplify the effect of OCBs on well-being at work, giving leadership a better-framed role. The leader does not have superpowers and is not responsible for all the good and bad that happens in organizations [187,188], rather the leader is one more important element in the complex network of relationships that are structured at work and who can contribute to more efficiency and well-being at work [189]. In fact, as our results indicate, more important than having good leaders is having good citizens.

### 5.1. Theoretical and Practical Contributions

The positioning of OCB as a predictor of well-being at work can change the focus of human resource management policies, which are increasingly concerned with valuing positive and collaborative environments (e.g., [4,14,190]), and may start to attribute greater importance to the ethical evaluation of candidates. The reframing of the importance of leadership in the organizational context and the evident ability of OCBs to generate more well-being at work opens up new possibilities for human resource management. The organizational culture, the work environment and the quality of social interactions, as factors that influence OCBs, assume greater importance in the search for valuing well-being at work and all the positive results associated with it. Faced with the conclusion that individual ethics can trigger different causality relationships that lead to more efficiency and well-being for organizations, ethical scrutiny, currently more concerned with monitoring behaviors after admission, could become one of the most important roles in the evaluation processes of new candidates to reinforce teams.

### 5.2. Limitations and Future Research

The present investigation has some limitations that should be mentioned. First, because we used convenience sampling, the results should be generalized with some caution concerning the Portuguese population and other countries. Another limitation is that the questionnaire was self-administered; somehow, the answers may induce some social desirability. A third limitation, the cross-sectional design of our study does not allow us to infer empirical causality among the constructs analyzed [191]. It is desirable that this study be replicated in the future by adopting a longitudinal design.

Organizational citizenship feeds several relevant organizational results, such as well-being at work, and is assumed to be a topic that deserves increasing attention from organizational research. Considering that our research model decomposes OCBs only into OCBI and OCBO, it will be relevant to deepen this decomposition for the five dimensions mentioned above and to assess, on the one hand, what is the contribution of each dimension to well-being at work and, on the other, how it interferes with the moderating effect of leadership styles. It will also be interesting to bring together new cultural and organizational contexts to validate the current hypotheses, as individual ethics, leadership and well-being at work are constructs that are greatly influenced by the specific contexts in which people work, in a network of interaction and conditioning that goes beyond the role they play or the organization where they work.

## Figures and Tables

**Figure 1 behavsci-13-00811-f001:**
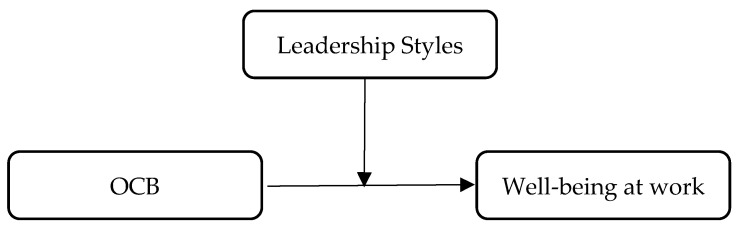
Hypothesized moderation model of leadership styles between OCBs and well-being at work.

**Figure 2 behavsci-13-00811-f002:**
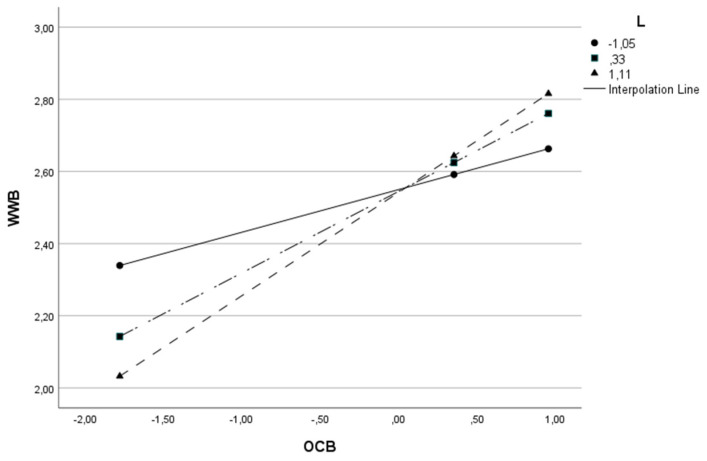
Interactive effect of Organizational Citizenship Behavior (OCB) and Leadership (L) on Work Well-Being (WWB).

**Figure 3 behavsci-13-00811-f003:**
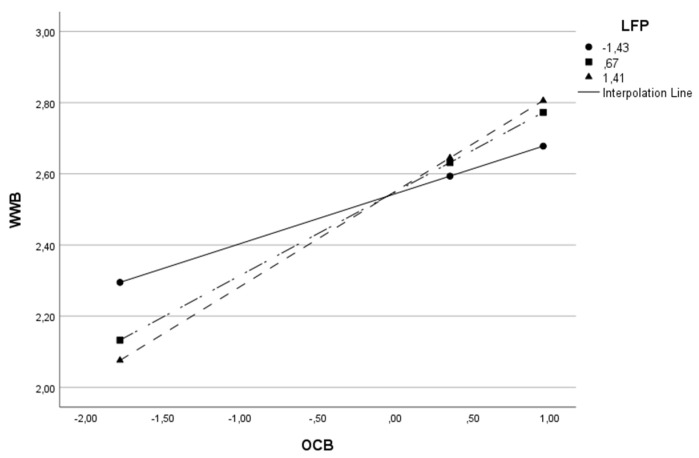
Interactive effect of organizational citizenship behaviors (OCB) and Leadership Focus on People (LFP) on Work Well-Being (WWB).

**Figure 4 behavsci-13-00811-f004:**
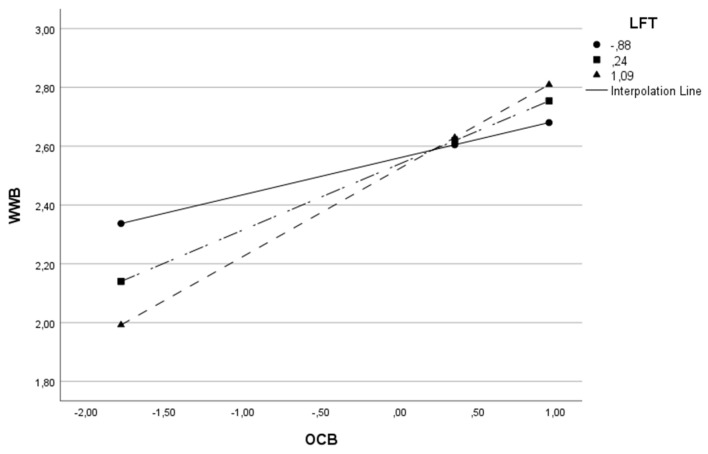
Interactive effect of organizational citizenship behaviors (OCB) and Leadership Focus on Task (LFT) on Work Well-Being (WWB).

**Figure 5 behavsci-13-00811-f005:**
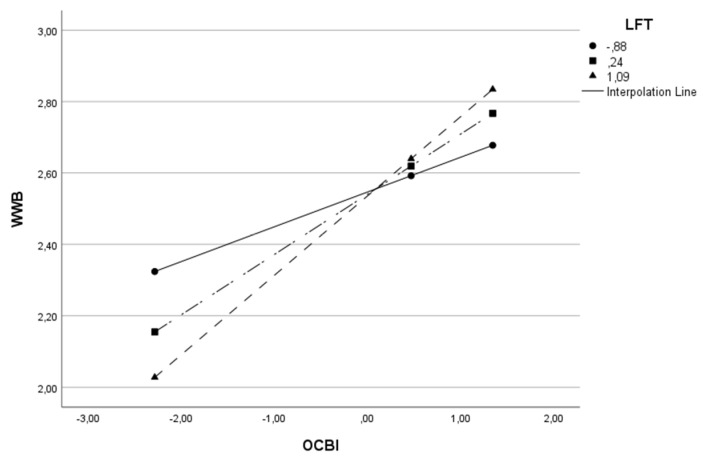
Interactive effect of individual-directed organizational citizenship behaviors (OCBI) and Leadership Focus on Task (LFT) on Work Well-Being (WWB).

**Figure 6 behavsci-13-00811-f006:**
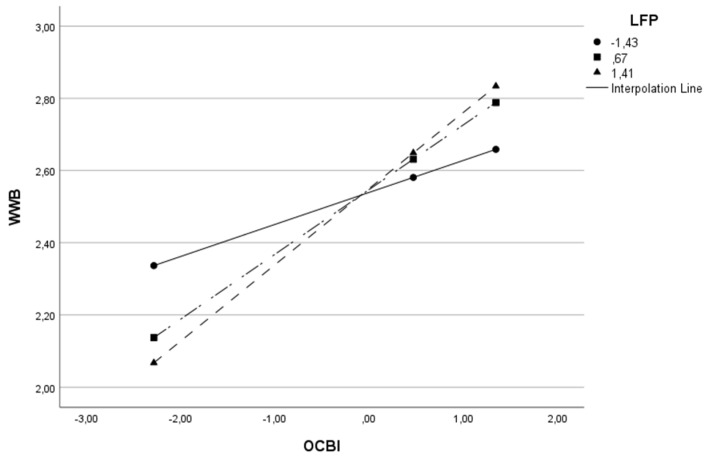
Interactive effect of individual-directed organizational citizenship behaviors (OCBI) and Leadership Focus on People (LFP) on Work Well-Being (WWB).

**Figure 7 behavsci-13-00811-f007:**
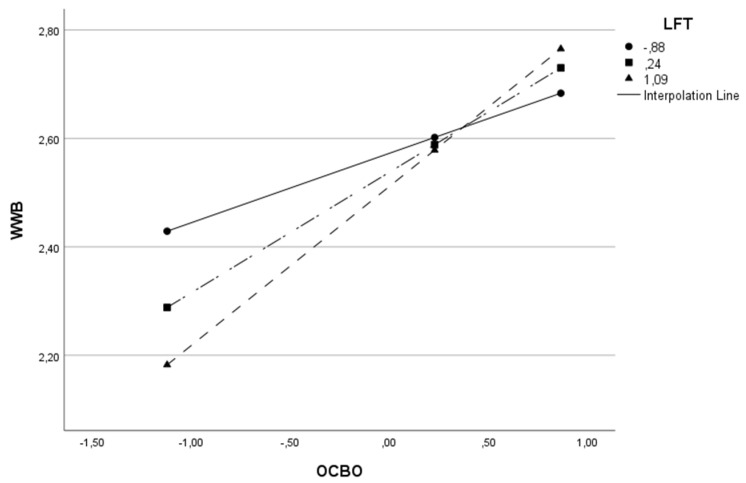
Interactive effect of organization-directed organizational citizenship behaviors (OCBO) and Leadership Focus on Task (LFT) on Work Well-Being (WWB).

**Table 1 behavsci-13-00811-t001:** Characterization of participants.

Participants	*n*	%	Mean ± SD
Gender			
Female	164	82.0	
Male	36	18.0	
Age (years)			40.0 ± 11.7
18–25	31	15.5	
26–35	43	21.5	
36–45	58	29.0	
46–55	46	23.0	
56–66	22	11.0	
Education Level			
Up to 12th grade	56	28.0	
Bachelor	94	47.0	
Master	44	22.0	
Missing	6	3.0	

**Table 2 behavsci-13-00811-t002:** Descriptive Statistics and Correlations for Scales and Subscales.

Scales and Dimensions		Mean	SD	Min	Max	1	1.1	1.2	2	2.1	2.2
1. Leadership		7.40	1.30	1.09	8.73						
	1.1 Task Focus	7.34	1.18	1.34	8.43	0.82 **					
	1.2. People Focus	7.46	1.77	0.16	9.02	0.92 **	0.54 **				
2. Organizational Citizenship Behavior		5.67	1.03	3.21	7.00	0.03	0.04	0.02			
	2.1 OCBI	5.66	1.46	2.00	7.00	0.04	0.05	0.02	0.94 **		
	2.2. OCBO	5.68	0.87	3.82	7.00	0.02	0.02	0.01	0.90 **	0.70 **	
3. Work Well-Being		2.55	0.44	1.31	3.95	0.02	−0.03	0.05	0.51 **	0.54 **	0.38 **

Note: Individual-directed organizational citizenship behaviors (OCBI); organization-directed organizational citizenship behaviors (OCBO); ** *p* < 0.001.

**Table 3 behavsci-13-00811-t003:** Moderation Effect of Leadership and Leadership styles on OCB, OCBI, OCBO and well-being at work.

Hypotheses		Coefficient (B)	SE	LL (95%)	UL (95%)	*t*	*p*
H3	Constant	2.546	0.026	2.495	2.597	98.023 ***	<0.0001
	OCB (X)	0.200	0.024	0.152	0.248	8.258 ***	<0.0001
	L (W)	−0.004	0.020	−0.044	0.036	−0.180	0.8575
	OCB × L (X×W)	0.078	0.021	0.037	0.118	3.781 **	0.0002
H3.a	Constant	2.544	0.026	2.495	2.595	98.010 ***	<0.0001
	OCB (X)	0.203	0.024	0.155	0.251	8.405 ***	<0.0001
	LFT (W)	−0.019	0.022	−0.062	0.025	−0.848	0.3972
	OCB × LFT (X×W)	0.088	0.023	0.042	0.135	3.768 **	0.0002
H3.b	Constant	2.547	0.026	2.495	2.599	96.913 ***	<0.0001
	OCB (X)	0.204	0.024	0.156	0.252	8.328 ***	<0.0001
	LFP (W)	0.002	0.016	−0.028	0.033	0.0150	0.8809
	OCB × LFP (X×W)	0.045	0.015	0.014	0.075	2.915 *	0.0040
H3.c	Constant	2.542	0.026	2.491	2.592	98.67 ***	<0.0001
	OCBI (X)	0.153	0.018	0.118	0.189	8.571 ***	<0.0001
	LFT (W)	−0.006	0.022	−0.050	0.038	−0.250	0.8030
	OCBI × LFT (X×W)	0.064	0.017	0.030	0.097	3.702 **	0.0003
H3.d	Constant	2.545	0.026	2.494	2.595	98.896 ***	<0.0001
	OCBI (X)	0.151	0.018	0.115	0.186	8.400 ***	<0.0001
	LFP (W)	0.004	0.015	−0.025	0.033	0.258	0.7965
	OCBI × LFP (X×W)	0.043	0.012	0.021	0.066	3.759 **	0.0002
H3.e	Constant	2.545	0.028	2.489	2.601	89.350 ***	<0.0001
	OCBO (X)	0.202	0.033	0.137	0.267	6.118 ***	<0.0001
	LFT (W)	−0.031	0.025	−0.080	0.018	−1.261	0.2087
	OCBO × LFT (X×W)	0.084	0.027	0.030	0.138	3.074 *	0.0024
H3.f	Constant	2.547	0.029	2.490	2.604	87.825 ***	<0.0001
	OCBO (X)	0.196	0.034	0.130	0.263	5.854 ***	<0.0001
	LFP(W)	0.005	0.017	−0.029	0.039	0.279	0.7808
	OCBO × LFP (X×W)	0.027	0.019	−0.010	0.064	1.416	0.1584

Note: Individual-directed organizational citizenship behaviors (OCBI); organization-directed organizational citizenship behaviors (OCBO); Leadership Focus on Task (LFT); Leadership Focus on People (LFP). * *p* < 0.1 ** *p* < 0.001 *** *p* < 0.0001.

## Data Availability

The data presented in this study are available on request from the corresponding author.

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
