# Peer review of "First, Be a Good Citizen: Organizational Citizenship Behaviors, Well-Being at Work and the Moderating Role of Leadership Styles"

_behavsci, 2023, doi:10.3390/bs13100811_

Round 1

Reviewer 1 Report

Dear Authors,

I find your text interesting and your methods of analysis adequate to the problem. The issue of explaining employee well-being is also socially important today. 

However, I see one key weakness in the project that causes me to suggest a reworking of the publication.

The problem is the lack of mention of the theory of demands - job resources Bakker and Demerouti (2017; 2018) Baker et al. 2023).

In JD-R theory, organizational behavior exemplified by OCBs is a consequence (!) of well-being and not the other way around. In the case of this text, citizenship behavior is treated as an antecedent of well-being (line 262/263). This is in clear contradiction to the available literature on the subject and needs to be discussed in the theoretical part. At the same time, the hypotheses only mention "relations." It needs to be unified whether it is a simple co-occurrence or cause-effect relationship. Regression analysis suggests that it is about directional relationship. Which indicates the inadequacy of the hypothesis for analysis.

In the text, the justification of the hypotheses (1.-1.2) is poorly supported by the literature - the passage introducing the hypotheses (lines 258-269) contains only one reference to the literature, and that not talking about the mechanism of the effect of OCB on well-being, but presenting the research tool. In order for the text to be published, it is necessary to discuss the question of what is the cause and what is the effect. All the more so since a cross-sectional study is presented here.

Author Response

Thank you very much for your suggestions for improvements. In the attached file you will find our answer and in the new version of the article the indication of the improvements introduced. Thanks.

Reviewer 2 Report

Main question addressed by the research: The Research investigates the relationship between employee behavior at work place  and their welfare in interraction with organizational leadership.                                                                                   

Originality and relevance of the topic including  research gap addressed: The topic is not only original but practically interactive,comparative,researchable and applicable in human resource management field.

What the topic  adds to the subject area compared with other published material: The Moderating role of leadership styles brought into the concept of people and task seeks to create new knowledge on employee welfare as a result of the interaction between  individual-directed  organizational citizenship and organizational-directed citizenship behaviours.

Specific improvements for the  authors to consider regarding the methodology and further controls suggestions. The authors should improve on the theoretical background ,hypothesis development and methodology by ;

Ø  drawing a visual conceptual framework showing the relationship of the variables involved

Ø  Specifying the sampling technique and procedure followed

Ø  Addressing the reliability of the data collection instruments

Ø  Discussing the validity of the research outcome and how it is representative and reproducible.

Are the conclusions consistent with the evidence and arguments presented and do they address the main question posed? The conclusions are consistent and remarkable.However,this needs further corroboration with recent research findings backed by relevant citations.

References. The authors should do a major revision of the references by replacing the old references with recent.(Preferably within the last five years).

Additional comments on the tables and figures. Consider including model summary and Clear ANOVA tables.

Plagiarism: 38.85 is too high, it must be reduced.

The authors should use past tense to show that their research has been concluded. For example in the abstract, they have to use

’’was ’’ instead of ’’will’’ (Line 12)

”Is” should be ”was’’(Line 56)

Also consider punctuation marks where necessary. 

Author Response

(The authors gave the same response as above.)

Round 2

Reviewer 1 Report

Dear Authors,

Thank you for your response and the changes you made to the text. It's a pity that you don't refer to the 2023 publication by Bakker, Demerouti and Sanz-Vegel. You mainly refer to a concpecation dating back to 2007. The feedback relationship is an interesting hypothesis, which in the 2017 and 2023 publications is better described as a positive and negative loop and always runs through resource modification. In this situation, interpreting OCB as a personality trait (personal resource) is an interesting thought. However, in the section on the description of the OCB variable, there is no mention of treating it as a trait. There is only a behavioral aspect.

In view of emphasizing the maneuverability of the OCB- Well Being relationship, a longitudinal study would be needed. At the moment, the title seems to me too boldly worded because the results of the correlation study do not prejudge the direction of the relationship. This should be stated in the limitations of the study.

The limitation also should take into account multiple explanations of the same variance pool (repeated consideration of the same explained variable in subsequent moderation analyses). It would be better to do a single regression analysis considering all moderators and their interactions.

Author Response

We appreciate the most recent comments, which contribute to improving the final version of the article. Thank you very much.

In the attached document, follow the respective answers.
